# Electrochemomechanical Behavior of Polypyrrole-Coated Nanofiber Scaffolds in Cell Culture Medium

**DOI:** 10.3390/polym11061043

**Published:** 2019-06-13

**Authors:** Madis Harjo, Janno Torop, Martin Järvekülg, Tarmo Tamm, Rudolf Kiefer

**Affiliations:** 1Intelligent Materials and Systems Lab, Faculty of Science and Technology, University of Tartu, Nooruse 1, 50411 Tartu, Estonia; madis.harjo@gmail.com (M.H.); jantor@ut.ee (J.T.); tarmo.tamm@ut.ee (T.T.); 2Institute of Physics, Faculty of Science and Technology, University of Tartu, W. Ostwaldi Str 1, 50411 Tartu, Estonia; martin.jarvekylg@ut.ee; 3Conducting polymers in composites and applications Research Group, Faculty of Applied Sciences, Ton Duc Thang University, Ho Chi Minh City 700000, Vietnam

**Keywords:** CFS, CFS-PPyTF, linear actuation, CCM, cation driven

## Abstract

Glucose-gelatin nanofiber scaffolds were made conductive and electroactive by chemical (conductive fiber scaffolds, CFS) and additionally electrochemical polypyrrole deposition (doped with triflouromethanesulfonate CF_3_SO_3_^−^, CFS-PPyTF). Both materials were investigated in their linear actuation properties in cell culture medium (CCM), as they could be potential electro-mechanically activated cell growth substrates. Independent of the deposition conditions, both materials showed relatively stable cation-driven actuation in CCM, based on the flux of mainly Na^+^ ions from CCM. The surprising result was attributed to re-doping by sulfate anions in CCM, as also indicated by energy-dispersive X-ray (EDX) spectroscopy results. Overall, the electrochemically coated material outperformed the one with just chemical coating in conductivity, charge density and actuation response.

## 1. Introduction

Tissue engineering materials can aid in repairing, retaining or improving tissue function [1]. In many cases electro-spun nanofiber scaffolds [2] from natural biomaterials such as cellulose [3], collagen [4] and gelatin [5] have been applied. Other materials such as conductive graphene composites [6,7] with hydroxyapatite forming hybrid bio-scaffolds composites have been shown to be effective in bone repair and regeneration [8]. Such materials can support and stabilize damaged muscle fibres [9]; after tissue has regenerated [10] they must degrade [1]. It has been demonstrated recently [11] that gentle agitation or actuation of such scaffolds can enhance healing. Such active nanofiber scaffolds could be adapted in smart patches. Additional smart functions (including sensing) can also be implemented, in case the scaffold is made conductive, for instance by including conducting polymers on the scaffold material [12,13]. Intensive studies of PPy deposited on poly-lactic acid fibres [12] as well conducting polymer surfaces have revealed direct influence on cardiac progenitor cells [14]. Further studies have shown that the delicate mechanical actuation originating from volume change can lead to the enhanced growth of epithelial cells [15].

There are certain limitations for using conducting polymers, since the scaffold material applied in tissue engineering can degrade in vivo, but the conducting polymer material is more durable [16]. This has led to the development of various composites and co-polymers with enhanced biodegradability. Conducting polymers are electroactive and can show reversible volume change (actuation) upon charging–discharging in an electrolyte [17]. Conducting polymers can be synthesized by either chemical oxidation using a strong oxidant such as ammonium persulfate (APS) forming solid particles [18] or deposited electrochemically on a conductive surface to obtain films. Chemical coatings based on PPy have been mostly applied for conductivity, electroactivity, with only few examples of actuation shown mainly for poly(3,4-ethylenedioxy-thiophene):poly(styrene sulfonate) (PEDOT:PSS) materials [19]. Electropolymerized conducting polymer films have usually been preferred for ionic actuators [20], where upon oxidation the PPy chains are charged and the counterions (with solvent molecules) migrate into the film leading to expansion (anion-driven actuators) [21]. During reduction, the PPy chains become neutral and the solvated counterions leave the film, leading to contraction. If large, bulky, multi-charged anions have been incorporated during polymerization, those anions often become immobile and the actuation during discharging is balanced by cation influx, which leads to film expansion upon reduction (cation-driven actuators) [22]. In particular cases even smaller anions such as CF_3_SO_3_^−^ can become (partially) entrapped in the conducting polymer film, resulting in mixed ion actuation [23].

The situation becomes more complex when an actuator is operated in another electrolyte solution, especially in one with a variety of ionic species, like biological fluids or culture media. As both the direction and the amplitude of actuation depend on the mobile species, the partial replacement of ions inside the film by those from the surrounding electrolyte may lead to the deterioration of the performance or inconsistent behavior, well known for un-encapsulated ionic actuators [24].

Therefore, our aim was to investigate whether the chemically and electrochemically deposited PPy coatings on nanofiber scaffolds would possess sufficient activity and stability of actuations in cell culture medium (CCM) in order to be used in potential smart patch or tissue engineering applications. The nanofiber scaffolds were based on glucose-crosslinked gelatin [25] with chemical PPy coatings (CFS, conductive nanofiber scaffolds [26]) with and without an additional electropolymerized PPy triflate (triflouromethanesulfonate, CF_3_SO_3_) (CFS-PPyTF) layer.

Isotonic and isometric Electro-Chemo-Mechanical-Deformation (ECMD) measurements of CFS and CFS-PPyTF were performed driven by cyclic voltammetry and chronoamperometry (frequencies 0.0025 Hz to 0.1 Hz) in the voltage range of 0.65 V to −0.6 V in CCM electrolyte to investigate the linear actuation response of the materials and to consider the suitability for biomedical applications. Characterization by Fourier transform infrared (FTIR) spectroscopy, scanning electron microscopy (SEM), electronic conductivity measurements and energy-dispersive X-ray (EDX) spectroscopy was carried out to evaluate the material properties and to understand the mechanisms of electro-mechanical behavior.

## 2. Material and Methods

### 2.1. Materials

Sodium dodecylbenzenesulfonate (NaDBS, technical grade), ammonium persulfate (APS, (NH_4_)_2_ technical grade) and tetrabutylammonium triflouromethanesulfonate (TBACF_3_SO_3_, 99%) and propylene carbonate (PC, 99%) were obtained from Sigma-Aldrich (Taufkirchen, Germany) and used as supplied. Pyrrole (Py, 98%, Sigma-Aldrich) was vacuum-distilled prior use and stored at low temperature in the dark. Milli-Q+ water was used for making aqueous solutions. Dulbecco’s modified Eagle’s medium (DMEM) with 4.5 g/L glucose content, penicillin/streptomycin (PENSTREP) solution and fetal bovine serum (FBS) was used as one of the electrolytes. Gelatin type A from porcine skin, D-(+)-glucose (99.5%) and glacial acetic acid (99%) used for producing the fibrous glucose-crosslinked gelatin were purchased from Sigma-Aldrich.

### 2.2. Electropolymerization of PPy Films on Conductive Fibre Scaffolds (CFS)

Fiber scaffolds based on glucose and type A gelatin from porcine skin were prepared by electrospinning, as described previously [26]. Obtained electrospun scaffolds were dipped in pyrrole monomer solution (2 M pyrrole in ethanol) followed by dipping several times in oxidant solution (0.075 M APS + 0.01 M NaDBS) to achieve conductive fiber scaffolds of thickness 49 ± 3 µm with electronic conductivity of 0.35 ± 0.02 S cm^−1^. The electrochemical PPy deposition was carried out on the CFS (galvanostatic 0.1 mA cm^−2^, at −20 °C, in 0.1 M pyrrole, 0.1 M TBACF_3_SO_3_ in propylene carbonate) for 40,000 s) in a two-electrode cell (CFS as the working electrode, stainless steel mesh as the counter electrode) resulting in CFS-PPyTF scaffolds. The CFS-PPyTF samples were washed with propylene carbonate and ethanol to remove excess of TBACF_3_SO_3_ and pyrrole, then dried in a vacuum oven at 60 °C (2 mbar) for 24 h. The thickness of the CFS-PPyTF samples was found in the range of 100 ± 5 µm.

### 2.3. Isometric and Isotonic ECMD Measurements

The CFS and CFS-PPyTF samples were cut into strips of 1.5 cm length and 0.1 cm width. In a three electrode set-up with a platinum counter electrode, Ag/AgCl (3M KCl) wire reference electrode, the samples were fixed on the gold contact/electrode of a force sensor (TRI202PAD, Panlab, Barcelona, Spain) in a linear muscle analyzer setup [27]. The in-house ECMD measurement with a movable force sensor allows us to determine the elasticity coefficient k (mg/ µm)– mass (mg) required to cause a 1 µm length change of a film; for CFS samples CCM the k_CFS_ was 2 mg/µm and for CFS-PPyTF samples the k_CFS-PPyTF_ was 116 mg/µm. The force (isometric, constant length of 1 mm), and the length change (isotonic, constant force of 4.9 mN) were measured in real time while running cyclic voltammetry (CV) with the scan rate of 5 mVs^−1^, within voltage range of 0.65 V to −0.6 V in an aqueous solution of CCM of DMEM with the addition of FBS and PENSTREP (500 mL of DMEM, 56 mL of FBS, and 4 mL of PENSTREP). Square wave potential steps of PPy-CFS samples were performed at 0.0025 Hz to 0.1 Hz. Stability measurements were performed for CFS-PPyTF samples at 0.1 Hz (600 cycles). For each measurement, three separate samples were taken, and the mean values with standard deviation of the results are presented.

### 2.4. Characterization of samples

The CFS and CFS-PPyTF (PPy doped with triflouromethanesulfonate) samples were characterized by scanning electron microscopy (Helios NanoLab 600, FEI, Hillsboro, Oregon, USA) and EDX (Oxford Instruments with X-Max 50 mm^2^ detector, Wiesbaden, Germany). Before characterization, 5 min polarization at −0.6 V and +0.65 V was performed for the reduced and oxidized films, respectively. To evaluate the surface conductivity of the coatings, an in-house 4-point probe was applied. Conductivity was calculated from Equation (1):(1)$$\sigma_{e}=\;\frac{1}{\left(R*w\right)}$$
where σ_e_ is the electric conductivity, R is the surface resistivity (Ω/sq) and w is the material thickness.

The change in the surface chemical composition of the scaffolds due to chemical and electrochemical coating was studied by FTIR spectroscopy (Bruker Alpha with Platinum ATR, Coventry, UK).

## 3. Results and Discussion

DMEM is designed to preserve and maintain the growth of a broad spectrum of mammalian cells, and it includes amino acids, vitamins and glucose. The inorganic salt content of the cell culture medium is shown in Table 1.

There are several of both anions and cations present, both single- and double-charged, however, Na^+^ and Cl^−^ are clearly dominant. This has to be kept in mind for the investigation of the mobile species participating in maintaining the electroneutrality during the redox cycling of the CFS and CFS-PPyTF samples.

### 3.1. Electropolymerization, Coating Morphology

The main reason for using organic electrolyte (TBACF_3_SO_3_ in PC) as the deposition medium instead of aqueous, relies on the nature of the CFS samples, which showed minor shrinking in propylene carbonate, which is favored to keep the conductive coatings intact. In case of aqueous solutions, the light swelling of the CFS samples could induce cracking of the chemically deposited PPy layer. The polymerization curve is shown in Figure 1a with the SEM images of CFS and CFS-PPyTF samples in Figure 1b,c.

The potential time curve (Figure 1a) of the galvanostatic electropolymerization revealed that the potential grew up to 3.4 V, and decreased thereafter to 3.2 V indicating that with PPyTF increasing deposition on CFS samples, conductivity (and surface area) increased. The conductivity of the CFS samples in the thickness of 48 ± 2 µm was in the range of 0.35 ± 0.02 S cm^−1^ [26] and the CFS-PPyTF samples in the thickness of 100 ± 5 µm had an electronic surface conductivity in range of 10 ± 0.52 S cm^−1^. The SEM surface images shown in Figure 1b of CFS revealed that the chemically deposited PPy was coating individual fibers, increasing their diameter from 0.8 µm [26] to about 1 µm. The typical cauliflower structure of PPyTF [28] was visible on the fibers of CFS-PPyTF. The fiber diameters after PPyTF deposition on CFS was about 1.5 µm (inset of Figure 1c). It is rather difficult to estimate the amount of PPyTF deposited, since the coating is relatively uneven and the scaffold itself has variable density and porosity.

As for the overall thickness increase of the fiber mats, it was found to be entirely due to PPy deposition. Swelling tests of dry thicker mats showed no thickness change (<1 μm) after immersion in CMC solution for 20 h, independent of the type of the mat (untreated fiber scaffold 48 µm, CFS 89 µm, CFS-PPyTF 145 µm). The uncoated cross-linked gelatin material did swell in range of 3% (up to 49–50.5 µm) when it was in contact with pyrrole monomer solution, but after the PPy coating surrounds the gelatin material, no further swelling was observed in CCM solution. Therefore, all thickness changes of PPy-coated scaffolds shown in this work can be attributed to the behavior of the deposited PPy and the electrochemically deposited PPyTF.

### 3.2. Elemental Composition

EDX spectroscopy of the CFS and CFS-PPyTF samples after actuation (200 cycles in CCM electrolyte) was performed to study the nature of the ionic species present upon oxidation (samples polarized at 0.65 V for 5 min) and reduction (−0.6 V for 5 min). The results are shown in Figure 2a,b.

Typical signals of carbon (C) at 0.27 keV, nitrogen (N) at 0.38 keV, oxygen (O) at 0.52 keV, sodium (Na) at 1.04 keV and sulfur (S) at 2.32 keV can be seen. The inset of Figure 2a,b shows two calcium peaks (Ca) at 3.72 keV and 4.02 keV. The CFS-PPyTF EDX spectrum in Figure 2b showed an additional small fluorine (F) peak at 0.68 keV. The spectra of the CFS demonstrated the influence of the redox state (oxidized vs. reduced) on the Na and Ca peaks. The originally incorporated SO_4_^2−^ counter-ions [18], products of the APS oxidizer, which compensate the positive charge of the as-formed PPy^n+^ are poorly mobile, due to their high charge density. This is illustrated here by the intensity of the sulfur peak, which appeared virtually independent of the redox state. Hence, the increase of the sodium (and calcium, to a lesser extent) peaks upon reduction, compensating for the negative charge of the entrapped sulfate anions, which is logical. Virtually the same behavior of the CFS-PPyTF samples (Figure 2b) is perhaps far more surprising. The electrodeposition was performed in TBACF_3_SO_3_, incorporating the triflate anions into the film, represented by the rather weak fluorine peak. The cation activity in CCM solution and the presence of a significant S peak can be explained by fast partial re-doping of the PPy material by the SO_4_^2−^ ions present in the CCM, which is known to take place even without redox cycling [29]. Since sulfate-doped films are typically more compacted and the dopant-polymer interactions strong (as sulfate “erases” the PPy memory of the original dopant [29]), it was understandable that anion activity was suppressed, there was even no evidence of chloride exchange, which was present in the CCM in a higher concentration. The low activity of the Ca^2+^ exchange was obviously related to the larger and stronger solvation shell, caused by the higher charge density. The lack of K^+^ participation must be related to the much lower concentration, as compared to Na^+^. Therefore, independent of the large variety of anions and cations present in the CCM, the charge neutrality was mostly established by the exchange of just one species – Na^+^. This result is both significant and advantageous, as it allowed us to expect to achieve the consistent controllability of electro-chemo-mechanical devices based on such materials in CCM solutions.

### 3.3. Fourier Transform Infrared (FTIR) Spectroscopy

To analyze the composition of the coated and uncoated scaffold materials, FTIR spectroscopy was performed. The results are shown in Figure 3.

Signals of fiber scaffolds (glucose-gelatin nanofiber scaffolds) in Figure 3 can be found at 3292 cm^−1^ representing the NH stretching vibration inherent to gelatin [30]. The 2939 cm^−1^ peak showed the CH_2_ asymmetric vibration of glucose gelatin material. The 1448 cm^−1^ peak showed different in plane vibrations [25]. The strong peak at 1640 cm^−1^ represented the amide I peak, the 1532 cm^−1^ the amide II and the 1238 cm^−1^ the amide III peaks [30,31]. The peaks at 3292 cm^−1^, 2939 cm^−1^, 1640 cm^−1^ and 1448 cm^−1^ of fiber scaffold are found as well in CFS and CFS-PPyTF materials (Figure 3). The CFS material showed one outstanding peak at 1782 cm^−1^, related to the C=O group attained due to over-oxidation [32] of the chemically deposited PPy by the APS as the oxidant [18]. Typical PPy signals [18,26] in Figure 3 of CFS and CFS-PPyTF are shown at 1540 cm^−1^ and 1484 cm^−1^, representing the C=C stretching modes. Additional peaks for PPy found at 1184 cm^−1^ (C–N stretching) and 1295 cm^−1^ (C–C stretching) can be detected only for CFS-PPyTF and the 1042 cm^−1^ peak represented the C–H in plane vibrations for the CFS. The 1027 cm^−1^ peak showed the dopant triflate anions [33] CF_3_SO_3_^−^ in CFS-PPyTF. Therefore, from FTIR analysis in Figure 3 it can be concluded that the peaks of neat unmodified scaffold material can still be found in the spectra of CFS and CFS-PPyTF samples. The typical PPy peaks found in CFS and CFS-PPyTF material confirmed the deposition, while the triflate anions can also be identified in the CFS-PPyTF samples.

### 3.4. Linear Actuation

#### 3.4.1. Cyclic Voltammetry

To investigate CFS and CFS-PPyTF samples in their linear actuation properties in CCM electrolyte, isotonic and isometric ECM measurements under cyclic voltammetry (0.65 V to −0.6 V, scan rate 5 mV s^−1^) were performed. The results are presented in Figure 4.

Figure 4a showed the strain and stress evolution of CFS samples in CCM electrolyte. Unexpectedly, even just the chemically deposited PPy coating on the nanofiber scaffold material exhibited a small linear strain in the range of 0.13% and a stress difference, obtained from the values upon oxidation and reduction, in the range of 0.32 kPa in CCM. This is not a typical result, as usually PPy deposited chemically and from a solution does not form coatings uniform enough to warrant actuation. The direction corresponded to cation-driven actuation, as expected from the EDX results (Figure 2a).

Cation-driven actuation was found also in case of CFS-PPyTF (Figure 4b), with increased strain in the range of 0.8% and stress in the range of 53 kPa.

As both types of materials were cation-active during the redox cycling, an Equation (2) can be formulated; where the left side corresponded to the reduced and the right side to the oxidized state.

(2)$$[{\left(PPy^{0}\right)}_{s}{\left(SO_{4}^{2-}\right)}_{n}{\left(2C^{+}\right)}_{n}{\left(S\right)}_{m}]⇄[\left(PPy^{2n+}\right){\left(SO_{4}^{2-}\right)}_{n}\left(S{)}_{m}\right]+2nC^{+}+2ne^{-}$$

The higher current density j against the applied potential E (potential range 0.65 V to −0.6 V) of the CFS-PPyTF samples compared to the CFS samples (Figure 4c) is explained by the increased thickness of the active material on the fibers as well as increased conductivity. While there were no clear redox peaks in either case, as expected [26], in the case of CFS-PPyTF, an oxidation wave could be detected at −0.05 V with a (narrower) reduction wave at −0.5 V. The location of the peaks in the negative potential region and the sharper shape of the reduction wave has also been previously related to the flux of cations rather than anions [34]. The closed loops of the charge density Q potential E curves of both systems in Figure 4d indicated that in the applied potential range charging/discharging was in balance [35]. The charge densities for CFS and for CFS-PPyTF were found in the range of 20 and 32 C cm^−3^, respectively.

#### 3.4.2. Step Response—Efficiency and Stability

Square wave potential step measurements combined with isotonic and isometric ECMD measurements were performed in the potential range 0.65 to −0.6 V in CCM electrolyte at 0.0025 Hz to 0.1 Hz to determine the redox charge density dependence on the frequency (a sample chronoamperogram of 0.0025 Hz is shown in Appendix A). Long-term isotonic ECMD measurements of CFS-PPyTF at 0.1 Hz (600 cycles) were performed to study the actuation stability, the results are shown in Figure 6.

From each CFS and CFS-PPyTF film, three different samples were measured. The mean values with standard deviations are shown in Figure 5b,c. The strain time curves at 0.01 Hz are shown in Figure 5a and the stress time curves in Appendix A.

Qualitatively, the step response results were in line with those of cyclovoltammetry—cation-driven actuation, with higher current densities (Appendix A), strain and stress for CFS-PPyTF than for CFS. The 0.1 Hz frequency was chosen as potentially suitable for the envisaged application in smart patches. As usual for ionic systems, both strain (Figure 5b) and stress (Appendix A) of CFS and CFS-PPyTF increased with decreasing driving frequency, reaching the maximum values at the lowest tested frequency 0.0025 Hz, which were strain 0.27% (stress 0.55 kPa, Appendix A) and 0.9% (114 kPa) for CFS and CFS-PPyTF, respectively. Appendix A shows the corresponding strain and stress values from each of the applied frequencies. It can to be noticed that both strain and stress) reached unmeasurable values by 0.1 Hz for CFS.

The higher current density shown by CFS-PPyTF was also more efficiently converted into actuation, as shown in Figure 5c by plotting strain against the reduction wave charge density (from the integration of the chronopotentiogram). The linear relationship between charge density and strain (or stress, Appendix A) is typical for a Faradaic actuator [36], the charge density determines the extent of actuation (strain and stress), but the slope is steeper for the electrochemically coated scaffold, which is in line with the common assumption that conducting polymers are more suitable actuator materials if electrodeposited rather than formed by chemical oxidation. In order to evaluate the stability and durability of the behavior of the electromechanically active scaffolds in CCM, 600 cycles of steps were applied at 0.1 Hz. The results for CFS-PPyTF are presented in Figure 6a,b.

While no creep could be observed during 600 cycles (6000 s) (Figure 6a), there was an obvious slight decrease in strain. The strain evolution and charge densities from cycle to cycle are shown in Figure 5b, revealing that the strain decreased from 0.16% to 0.12% (a decrease in range of 25%) as the charge density dropped from −0.18 C cm^−3^ to −0.14 C cm^−3^. For gentle agitation in a cell-growth application scenario, with longer pauses between motions, the stability can even be increased. In any case, the decrease in amplitude appeared to continue and does not indicate abrupt changes caused by redoping or other medium-related issues.

In all respects, the actuation performance of CFS-PPyTF samples in the CCM solution was found to be higher than that of CFS samples, with higher current and charge densities, strain and stress. Therefore, in the case of medical applications, the agitation-induced increased cell-growth rate (healing) could be higher for CFS-PPyTF. However, the actual influence on cells needs to be determined for both materials, as other aspects like hydrophilicity/hydrophobicity, surface morphology, porosity, etc. are at play there in addition to mechanical stimulus. The cytotoxicity of chemically polymerized PPy has been investigated before [37]. There was some indication of cytotoxicity at higher concentrations, however, in the case of PPy the results strongly depend on various parameters, including the redox state and the dopant ions, therefore, the topic deserves a more detailed analysis The interaction of PPy with cells as well their cytotoxicity and biodegradability will be investigated in future work.

## 4. Conclusions

Nanofiber scaffold materials of gelatin-crosslinked glucose coated with chemically deposited PPy (CFS) were further coated with electrochemically deposited PPyTF (CFS-PPyTF) characterized by FTIR measurements. In both cases the PPy covered individual fibers without filling the voids in between. The two materials were characterized as potential electro-mechanically active cell substrates in cell-growth medium. While deposited differently and (initially) containing different dopant anions, both electroactive coatings showed cation-activity based on the flux of mainly solvated Na^+^ (and Ca^2+^ to a lesser extent) from all the possible ions presented in CCM solution. This somewhat surprising result was attributed to redoping of the electrodeposited PPy by the sulfate anions from CCM, supported by the stable sulfur peak in EDX. Accordingly, both materials showed surprisingly stable actuation performance considering the complicated conditions. This stable behavior is very encouraging, as consistent actuation in a mixed electrolyte solution, where both small anions and cations are present, is a prerequisite for attempting gentle mechanical activation of biomaterials in medical applications. In each of the experiments, the electrochemical PPy layer of CFS-PPyTF led to increased performance over CFS, in terms of current density, charge efficiency, strain and stress. Further investigation are in progress to test CFS and CFS-PPyTF materials in cell culture tests to explore the possibility of in vivo micro-mechanical stimulation of mammalian cells.

## Figures and Tables

**Figure 1 polymers-11-01043-f001:**
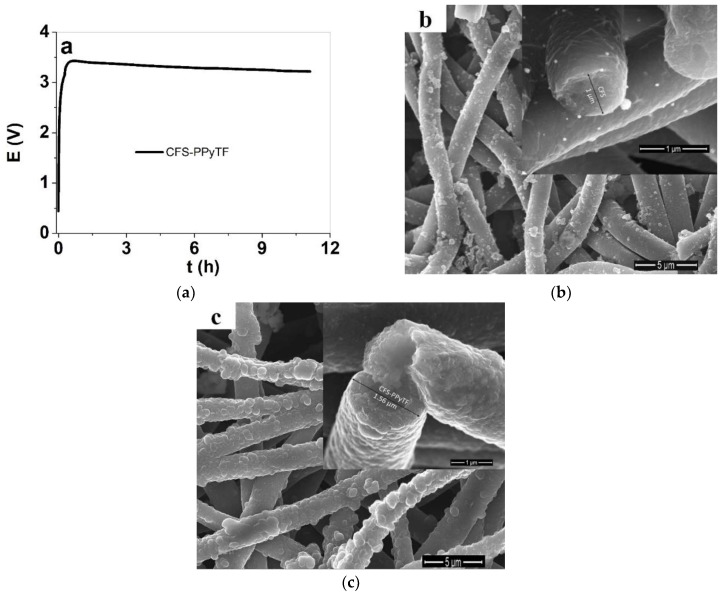
(**a**): Galvanostatic electropolymerization curve (0.1 mA cm^−2^, −20 °C, 40,000 s, 0.1 M TBACF_3_SO_3_, 0.1 M Py, propylene carbonate) in two electrode cell (counter electrode stainless steel mesh and working electrode conductive fiber scaffolds (CFS)) of CFS-PPyTF. The scanning electron microscope (SEM) images (scale bar 5 µm) with inset of single fiber (scale bar 1 µm) of (**b**) CFS and (**c**) CFS-PPyTF.

**Figure 2 polymers-11-01043-f002:**
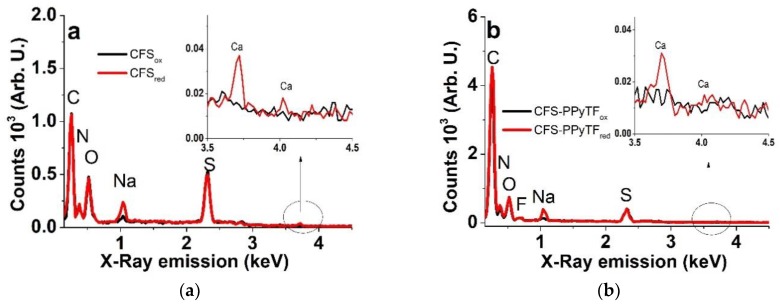
Energy-dispersive X-ray (EDX) spectra of oxidized (5 min, 0.65 V, black line) and reduced (5 min at −0.6 V, red line) cross sections of samples of (**a**) CFS and (**b**) CFS-PPyTF with insets showing calcium evolution.

**Figure 3 polymers-11-01043-f003:**
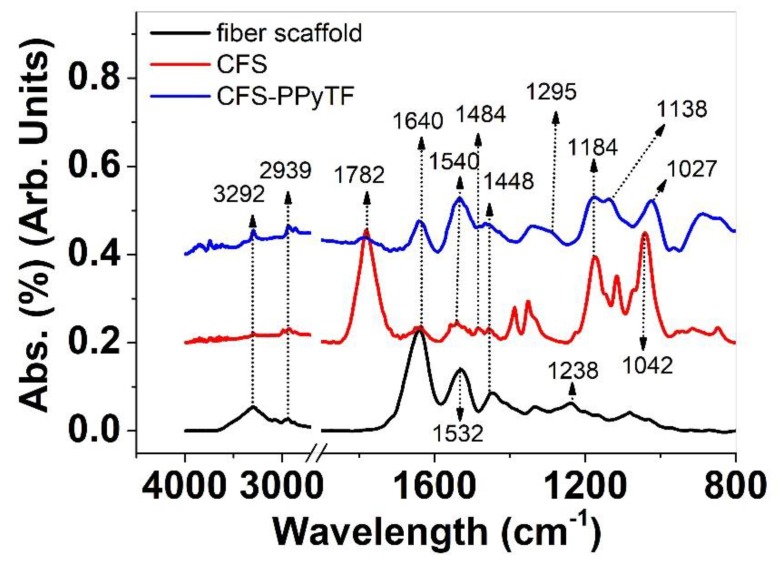
Fourier transform infrared (FTIR) spectra (4000–800 cm^−1^) of fiber scaffold (black), CFS (red) and CFS-PPyTF (blue).

**Figure 4 polymers-11-01043-f004:**
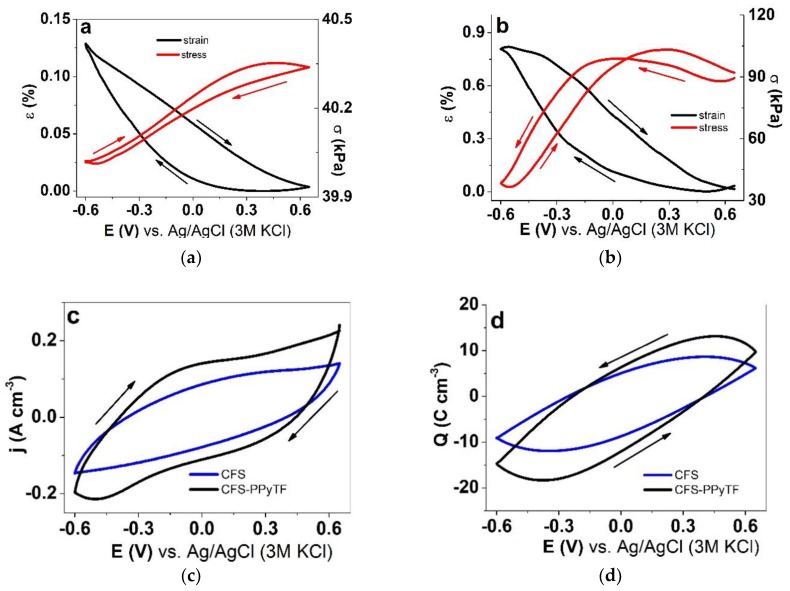
Cyclic voltammetry (scan rate 5 mV s^−1^) of CFS and CFS-PPyTF samples in cell culture medium (CCM) electrolyte (potential range 0.65 V to −0.6 V, 4th cycle) showing in strain ε (black) and stress σ (red) of (**a**) CFS samples strain ε and stress σ against potential E; (**b**) CFS-PPyTF samples strain ε and stress σ against potential E. The current density j and charge density curves Q against potential E of CFS samples (blue) and CFS-PPyTF samples (black) are shown in (**c**) and (**d**), respectively. The arrows indicate the start and end of the cycles.

**Figure 5 polymers-11-01043-f005:**
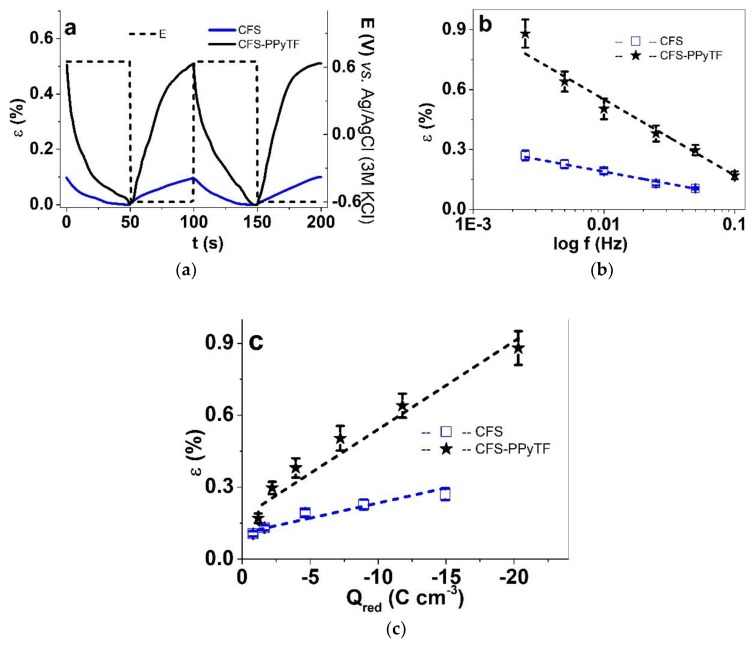
Square wave potential steps response in potential range 0.65 V to −0.6 V in CCM electrolyte showing in (**a**) the strain ε curves (3rd–5th cycles) of CFS (blue) and CFS-PPyTF (black) with potential E (dashed) against time t (frequency, 0.01 Hz). At frequencies 0.0025 Hz to 0.1 Hz the strain of CFS (blue, □) and CFS-PPyTF (black, ★) against logarithm of frequency are shown in (**b**). The strain ε against the charge density Q_red_ is shown in (**c**). The dashed lines in b and c represent the linear fit (y = a + b*x, with adj. R^2^ of 0.91 for CFS-PPyTF and 0.93 for CFS), shown as visual guides.

**Figure 6 polymers-11-01043-f006:**
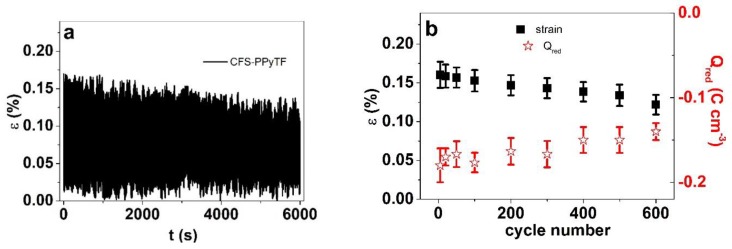
Square wave potential steps at 0.1 Hz (600 cycles) in potential range 0.65 V to −0.60 V in CMC solution of CFS-PPyTF samples showing in (**a**) the strain ε against time and (**b**) the strain ε and charge density at reduction Q_red_ against cycle number.

**Table 1 polymers-11-01043-t001:** Ion content of the Dulbecco’s modified Eagle’s medium (DMEM) cell culture medium.

Inorganic Salts	Concentration [g L^−1^]
NaHCO_3_	3.7
NaCl	6.4
CaCl_2_	0.2
MgSO_4_	0.097
NaH_2_PO_4_	0.109
KCl	0.4

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
