# Peer review of "Electrochemomechanical Behavior of Polypyrrole-Coated Nanofiber Scaffolds in Cell Culture Medium"

_polymers, 2019, doi:10.3390/polym11061043_

Reviewer 1 Report

Title: Electrochemomechanical behavior of polypyrrole-coated nanofiber scaffolds in cell culture medium
Authors: Madis Harjo, Janno Torop, Martin Jarvekulg, Tarmo Tamm and Rudolf Kiefer
Manuscript ID: polymers-515817

The submitted work demonstrate a study of conductive and electroactive glucose-gelatin nanofiber scaffolds coated with polypyrrole. The resubmitted work was improved by adding some more details about the fiber morphologies before and after deposition of polypyrrole on the fiber surfaces as well as the FTIR investigations clearly demonstrated the effect of the polypyrrole treatment. The resubmitted work is recommended for publication in Polymers after minor revision.

Comments and suggestions:
1) in the Introduction part the sentence in line 61 has no verb, the referee thinks it refers to the materials which were used for the study. The authors should check the paragraph
2) the last paragraph in the Introduction part is more experimental related, so it should be rephrased more to show the aim and what was wanted to achieve with this work a small overview.
3) the typos in the manuscript should be checked and corrected, e.g. line 85 "pyrrol" should be changed to "pyrrole"
4) the authors should rephrase the sentence in line 153, because "somewhat" is not an exact definition. The gravimetric or volumetric swelling should be demonstrated with data to compare the results before and after coating.

Author Response

Reviewer 1

We thank the reviewer for the positive comments regarding our manuscript and have revised it as suggested.

Title: Electrochemomechanical behavior of polypyrrole-coated nanofiber scaffolds in cell culture medium
Authors: Madis Harjo, Janno Torop, Martin Jarvekulg, Tarmo Tamm and Rudolf Kiefer
Manuscript ID: polymers-515817

The submitted work demonstrate a study of conductive and electroactive glucose-gelatin nanofiber scaffolds coated with polypyrrole. The resubmitted work was improved by adding some more details about the fiber morphologies before and after deposition of polypyrrole on the fiber surfaces as well as the FTIR investigations clearly demonstrated the effect of the polypyrrole treatment. The resubmitted work is recommended for publication in Polymers after minor revision.

Comments and suggestions:
1) in the Introduction part the sentence in line 61 has no verb, the referee thinks it refers to the materials which were used for the study. The authors should check the paragraph

We have corrected this term

2) the last paragraph in the Introduction part is more experimental related, so it should be rephrased more to show the aim and what was wanted to achieve with this work a small overview.

The last to paragraphs of the introduction were revised to clarify the aims of the research.

3) the typos in the manuscript should be checked and corrected, e.g. line 85 "pyrrol" should be changed to "pyrrole"

We thank the reviewer for pointing this out and have corrected typographic errors in that particular phrase and elsewhere.

4) the authors should rephrase the sentence in line 153, because "somewhat" is not an exact definition. The gravimetric or volumetric swelling should be demonstrated with data to compare the results before and after coating.

We thank the reviewer and rephrased the paragraph about swelling, also including numeric results.

As for the overall thickness increase of the fiber mats, it was found to be entirely due to PPy deposition. Swelling tests of dry thicker mats showed no thickness change (< 1 mm) after immersion in CMC solution for 20 hours, independent of the type of the mat (untreated fiber scaffold 48 µm, CFS 89 µm, CFS-PPyTF 145 µm). The uncoated cross-linked gelatin material did swell in range of 3 % (up to 49-50.5 µm) when it was in contact with pyrrole monomer solution, but after the PPy coating surrounds the gelatin material, no further swelling was observed in CCM solution. Therefore, all thickness changes of PPy-coated scaffolds shown in this work can be attributed to the behavior of the deposited PPy and the electrochemical deposited PPyTF.

Reviewer 2 Report

Review comments

In this manuscript, the authors reported glucose-gelatin nanofiber scaffolds with electrochemical polypyrrole deposition and the materials are investigated in their linear actuation properties in cell culture medium. This reviewer recommends     this manuscript to be accepted after the following questions are properly addressed:
1. Please adjust the format of this paper. All the figures and equations should satisfy the standard of the journal, spacing is required between numbers and units……

2. Please increase some characterization of the materials.

3. Please increase the graph of the process flow chart of the experiment and the pictures of the materials

4. If possible, please provide the Cytotoxicity of the materials?

5. Please note the tenses in the paper.

6. Please explain why choose 40000 s in the electrochemical PPy deposition, if possible, please provide some parameters of other time?

7. Please explain the necessity of galvanostatic electropolymerization curve in figure 1.

8. Figure 4 is incorrectly formatted and the coordinate axis is incomplete. 4C and 4D lack the explanation of coordinate axis.

9. Please explain why the stresses in figs. 4A and B do not start from 0 and the strains in 4A do not start from 0.

10. Please explain why only the fourth cycle is selected in Figure 4 and why the third to fifth cycle is selected in Figure 5. It is suggested to add more cyclic data or give reasonable explanations.

11. Line 224, Figure 4A does not show where the 0.32 kPa stress of CCM is.

12. Line 224-225 "This is not a typical result, as usual PPy, deposited chemically and from a solution," has problems with grammar

13. Line 240, "The closed loops of the charge density time curves of both systems in Figure 4 d" There is no variable“time” in Figure 4 D

14. The illustration in Figure 5 needs to be consistent with the curve on the color and line type and mark.

15. Please further prove whether the variables in Fig. 5B and Fig. 5C conform to the law of linear variation. When using linear fitting, please give the fitting formula and goodness of fit.

16. As can be seen from Fig. 5a, the strain is also changing in a potential period. Please explain which value of the strain given in Fig. 5b is from each period.

17. These publications should be further reviewed in the introduction section, e.g., Crystals 8 (2), 105:1-12; Carbon 124 (C), 296-307; Advanced Materials 29 (28), 1605506.

Author Response

Reviewer 2 

We thank the reviewer for the useful comments and have revised the manuscript

In this manuscript, the authors reported glucose-gelatin nanofiber scaffolds with electrochemical polypyrrole deposition and the materials are investigated in their linear actuation properties in cell culture medium. This reviewer recommends this manuscript to be accepted after the following questions are properly addressed:

1. Please adjust the format of this paper. All the figures and equations should satisfy the standard of the journal, spacing is required between numbers and units……

Thank you for pointing, we have corrected all terms of spacing, figures and numbers and units

2. Please increase some characterization of the materials.

Since most of the background characterization of the underlying fibre scaffolds as well as the chemical PPy coating has been already published previously, our intention here was to conserve space and keep the focus on the ion exchange in cell culture medium and actuation behaviour. A more detailed characterization of the material is given in reference 22:

Harjo, M.; Kesküla, A.; Leemets, K.; Khorram, M.S.; Saar, R.; Järvekülg, M.; Tamm, T.; Kiefer, R. Polypyrrole coatings on gelatin fiber scaffolds: Material and electrochemical characterizations in organic and aqueous electrolyte. Synth. Met. 2017, 232, 25–30.

3. Please increase the graph of the process flow chart of the experiment and the pictures of the materials

We have increased the resolution of the flow charts and the SEM images

4. If possible, please provide the Cytotoxicity of the materials?

Good point. However, cytotoxicity experiments were not in the scope of the present manuscript, as this issue deserves special attention in the future. There have been only a few studies presented in the literature, there is one indicating that chemically formed  PPy cytotoxicity is dose dependent - against Jurkat, MEF and MH-22A cells. A sentence has been included with the new reference [37] in last section of the manuscript.

The cytotoxicity of chemically polymerized PPy has been investigated before[37]. There was some indication of cytotoxicity at higher concentrations, however, in case of PPy the results strongly depend on various parameters, including the redox state and the dopant ions, therefore, the topic deserves a more detailed analysis The interaction of PPy with cells as well their cytotoxicity and biodegradability will be investigated in future work.

5. Please note the tenses in the paper.

We have corrected the tenses in several places of the manuscript

6. Please explain why choose 40000 s in the electrochemical PPy deposition, if possible, please provide some parameters of other time?

At constant current density, the deposition time corresponds to the deposited polymer thickness. For various synthesis conditions, estimates are available in the literature. 40,000 s provided sufficient thickness of the polymer, however, the synthesis conditions can surely be further optimized.

7. Please explain the necessity of galvanostatic electropolymerization curve in figure 1.

Galvanostatic regime has been the favourite electrochemical deposition method for PPy ionic actuator and sensor fabrication, as constant current density ensures uniform polymer growth rate (and more or less uniform properties).  The curve describes the voltage evolution; any abnormalities would indicate a) loss of conductivity; b) increase or decrease of active surface area, etc. In our case, the PPy deposited increases the conductivity as well as slightly the diameter of the CFS fibers, therefore the potential is lowered gradually.  For some readers, it would surely provide information they look for.

8. Figure 4 is incorrectly formatted and the coordinate axis is incomplete. 4C and 4D lack the explanation of coordinate axis.

We thank the reviewer and have correct Figure 4a, b and have explained Figure 4c and d the explanation of the axis

9. Please explain why the stresses in figs. 4A and B do not start from 0 and the strains in 4A do not start from 0.

 Strain starts from 0 in first cycle, corresponding to initial state at 0V but here are the fourth cycle data in potential range 0.65 to -0.6 V. In case of stress there no 0 values because the samples are under slight pretension before measurement starts, to ensure comparability.

10. Please explain why only the fourth cycle is selected in Figure 4 and why the third to fifth cycle is selected in Figure 5. It is suggested to add more cyclic data or give reasonable explanations.

Usually the 1st and often the 2nd cycles are omitted from the results of ion-active materials due to irreversible and often unreproducible “break in” processes, which depend on the often difficult-to-control preconditions. Some authors even cycle longer before they consider their material has reached “steady state” to produce consistent results.  It is, naturally, arbitrary and depends on the point to be made and the particular materials.

11. Line 224, Figure 4A does not show where the 0.32 kPa stress of CCM is.

The stress curve shows a contraction at reduction (lower stress with reduction, increase with oxidation). The difference between s at reduction and oxidation gives 0.32 kPa, corresponding to the force the material exhibits to the clamps during a cycle – the force can be employed for generating work. Additional explanation was inserted in page 7, line 235-236.

12. Line 224-225 "This is not a typical result, as usual PPy, deposited chemically and from a solution," has problems with grammar

This is corrected now

13. Line 240, "The closed loops of the charge density time curves of both systems in Figure 4 d" There is no variable“time” in Figure 4 D

We have changed this to potential and thank the reviewer for pointing out our mistake.

14. The illustration in Figure 5 needs to be consistent with the curve on the color and line type and mark.

Figure 5 has been modified

15. Please further prove whether the variables in Fig. 5B and Fig. 5C conform to the law of linear variation. When using linear fitting, please give the fitting formula and goodness of fit.

Explanation given in Figure caption 5

The dashed lines in b and c represent the linear fit (y = a + b*x, with adj. R square of 0.91 for CFS-PPyTF and 0.93 for CFS), shown as visual guides.

16. As can be seen from Fig. 5a, the strain is also changing in a potential period. Please explain which value of the strain given in Fig. 5b is from each period

We have included a new Table S1 in supplementary which shows the strain e and stress difference Ds values at each applied frequency of CFS and CFS-PPyTF samples shown in Figure 5b and Figure S2a.

17. These publications should be further reviewed in the introduction section, e.g., Crystals 8 (2), 105:1-12; Carbon 124 (C), 296-307; Advanced Materials 29 (28), 1605506.

References have been included in the introduction

Other materials such as conductive graphene composites[6,7] with hydroxyapatite forming hybrid bio-scaffolds composites has been shown being effective in bone repair and regeneration [8].